# Species Diversity, Biomass Production and Carbon Sequestration Potential in the Protected Area of Uttarakhand, India

**DOI:** 10.3390/plants14020291

**Published:** 2025-01-20

**Authors:** Geetanjali Upadhyay, Lalit M. Tewari, Ashish Tewari, Naveen Chandra Pandey, Sheetal Koranga, Zishan Ahmad Wani, Geeta Tewari, Ravi K. Chaturvedi

**Affiliations:** 1Department of Botany, Kumaun University, D.S.B. Campus, Nainital 263001, Uttarakhand, India; geetanjaliupadhyay795@gmail.com (G.U.); pandeynaveen10@gmail.com (N.C.P.); sheetalkoranga98@gmail.com (S.K.); 2Department of Forestry and Environment, Kumaun University, D.S.B. Campus, Nainital 263001, Uttarakhand, India; atewari69@gmail.com; 3Terrestrial Ecology and Modelling (TEaM), Department of Environmental Science and Engineering, SRM University-AP, Amravati 522240, Andha Pradesh, India; zishanwani786@gmail.com; 4Department of Chemistry, Kumaun University, D.S.B. Campus, Nainital 263001, Uttarakhand, India; 5Center for Integrative Conservation, Xishuangbanna Tropical Botanical Garden, Chinese Academy of Sciences, Menglun, Mengla, Jinghong 666303, China

**Keywords:** carbon sequestration, correlation, forest management, restoration, species richness, Binsar Wildlife Sanctuary

## Abstract

Ecosystem functioning and management are primarily concerned with addressing climate change and biodiversity loss, which are closely linked to carbon stock and species diversity. This research aimed to quantify forest understory (shrub and herb) diversity, tree biomass and carbon sequestration in the Binsar Wildlife Sanctuary. Using random sampling methods, data were gathered from six distinct forest communities. The study identified 271 vascular plants from 208 genera and 74 families. A notable positive correlation (r^2^ = 0.085, *p* < 0.05) was observed between total tree density and total tree basal area (TBA), shrub density (r^2^ = 0.09), tree diversity (D) (r^2^ = 0.58), shrub diversity (r^2^ = 0.81), and tree species richness (SR) (r^2^ = 0.96). Conversely, a negative correlation was found with the concentration of tree dominance (CD) (r^2^ = 0.43). The *Quercus leucotrichophora*, *Rhododendron arboreum* and *Quercus floribunda* (QL-RA-QF) community(higher altitudinal zone) exhibited the highest tree biomass (568.8 Mg ha^−1^), while the (*Pinus roxburghii* and *Quercus leucotrichophora*) PR-QL (N) community (lower altitudinal zone) in the north aspect showed the lowest (265.7 Mg ha^−1^). Carbon sequestration was highest in the *Quercus leucotrichophora*, *Quercus floribunda* and *Rhododendron arboreum* (QL-QF-RA) (higher altitudinal zone) community (7.48 Mg ha^−1^ yr^−1^) and lowest in the PR-QL (S) (middle altitudinal zone) community in the south aspect (5.5 Mg ha^−1^ yr^−1^). The relationships between carbon stock and various functional parameters such as tree density, total basal area of tree and diversity of tree showed significant positive correlations. The findings of the study revealed significant variations in the structural attributes of trees, shrubs and herbs across different forest stands along altitudinal gradients. This current study’s results highlighted the significance of wildlife sanctuaries, which not only aid in wildlife preservation but also provide compelling evidence supporting forest management practices that promote the planting of multiple vegetation layers in landscape restoration as a means to enhance biodiversity and increase resilience to climate change. Further, comprehending the carbon storage mechanisms of these forests will be critical for developing environmental management strategies aimed at alleviating the impacts of climate change in the years to come.

## 1. Introduction

Globally, natural reserved forests serve as habitats rich in biodiversity, housing a wide array of plant and animal species [1,2]. As a crucial element of the natural world, forests play a key role in stabilizing the climate [3]. They serve multiple functions: regulating ecosystems, safeguarding biodiversity, playing an essential part in the carbon cycle, supporting human livelihoods and offering goods and services that contribute to sustainable development [4]. Human activities pose a significant threat to forests [5]. The primary factors endangering biodiversity and the environment include population growth, agricultural expansion, intensification and infrastructure development, which contribute to increased atmospheric CO_2_ levels [6,7,8]. Evaluating forest community composition and structure is crucial for comprehending tree population dynamics and diversity [9,10]. Himalayan forests, typically governed by low temperatures, are particularly vulnerable to the pronounced effects of climate change [11]. Comparing current vegetation with historical patterns allows for tracking ongoing changes in Himalayan forests. Consequently, quantitative analysis of these forests is essential to evaluate climate change’s impact on future species coexistence, establish baseline data for long-term monitoring and assess species shifts [10,12].

Moreover, issues such as global climate change, habitat fragmentation, land degradation, land-use alterations and sudden canopy gaps due to intensive logging significantly affect forest tree populations and their diversity [2,13,14,15]. This is particularly relevant in countries like India, where forests play a crucial role in rural communities’ food security [16,17,18]. Situated in the foothills of the Himalayan Mountain ranges, Uttarakhand State constitutes the central portion of the Indian Himalayan Region (IHR). The state is also distinguished by its significant forest carbon reserves, ranking fourth in India with 62.77 Mg ha^−1^ of aboveground biomass [7]. An investigation by Tolangay and Moktan (2020) revealed that the Indian Himalayan region (IHR) alone sequesters 65 million tons of carbon annually, emphasizing the potential of Himalayan forests in mitigating climate change and addressing global warming [19]. Numerous studies underscore the importance of assessing structure, diversity, carbon storage and sequestration in various protected areas, including wildlife sanctuaries [8,20,21,22,23,24,25,26,27]. Protected areas play a key role in carbon (C) sequestration by rapidly accumulating carbon [28,29], thus significantly contributing to global warming mitigation [30]. Previous studies in the Binsar Wildlife Sanctuary primarily focused on the phytosociological characteristics of tree species [27,31,32]. Limited research has been conducted on other life forms, such as shrubs and herbs [33,34]. Furthermore, Joshi et al. (2024) have examined the carbon stock of the trees in this study site [27]. Therefore, the current research aims to quantify forest understory (shrub and herb) diversity, tree biomass and carbon sequestration potential of the forests in the Binsar Wildlife Sanctuary (BWLS), Uttarakhand, India, to find the answers tothe following questions:(1) How does density and total basal area vary in different forest communities? (2) How much carbon do different forest communities sequester annually? (3) Do altitudes influence the tree density, biomass and carbon sequestration potential of forests? (4) What is the relationship between vegetation attributes and carbon stock?

## 2. Results

### 2.1. Phytocoenology Study

The current study identified 271 vascular plant species from 208 genera and 74 families. The flora comprised 214 herbaceous plants, 31 shrubs and 26 trees. Four gymnosperm species were recorded, namely *Picea smithiana*, *Cupressus torulosa*, *Cedrus deodara* and *Pinus roxburghii*. Asteraceae emerged as the most dominant family with 38 species, followed by Poaceae (21 species), Fabaceae (18 species), Lamiaceae (17 species), Rosaceae (10 species), Apiaceae, Cyperaceae, and Orchidaceae (7 species each) (Figure 1). Of the 74 recorded families, 29 were represented by a single species namely Anacardiaceae, Apocynaceae, Aquifoliaceae, Begoniaceae, Betulaceae, Brassicaceae, Celasteraceae, Corariaceae, Cucurbitaceae, Cupressaceae, Daphniphyllaceae, Droseraceae, Euphorbiaceae, Hypericaceae, Linaceae, Lythraceae, Malvaceae, Menispermaceae, Moraceae, Myricaceae, Myrtaceae, Onagraceae, Orobanchaceae, Phyllanthaceae, Rutaceae, Sapindaceae, Saxifragaceae, Scrophulariaceae and Thymeleaceae. *Geranium* and *Erigeron* were the dominant genera with four species (Figure 2). *Pinus roxburghii* and *Quercus leucotrichophora* (PR-QL) mixed community was recorded in both south and north aspects at 1600–1900 m and 1900–2100 m altitudinal range. *Pinus roxburghii* (PR) community was recorded at an elevation range of 1600–1900 m in the south aspect, *Quercus leucotrichophora* and *Rhododendron arboreum* (QL-RA) mixed community was found ranging between 1900 and 2100 in north aspect; *Quercus leucotrichophora*, *Rhododendron arboreum* and *Quercus floribunda* (QL-RA-QF) mixed community was mostly abundant between 2100 and 2400 m in north aspect, whereas, *Quercus leucotrichophora*, *Quercus floribunda* and *Rhododendron arboreum* (QL-QF-RA) mixed community was recorded between 2100 and 2400 m elevation in south aspect.

Among the identified forest communities, the minimum and maximum number of trees were recorded in the PR community (7) and QL-RA-QF (18). The overall tree density within the identified forest communities ranged from 663 to 1066 individuals ha^−1^ and the total basal area from 40.24 to 71.20 m^2^ ha^−1^ (Figure 3). A significant positive correlation (r^2^ = 0.085, *p* < 0.05) was found between total tree density and total tree basal area, shrub density (r^2^ = 0.09), tree diversity (r^2^ = 0.58), shrub diversity (r^2^ = 0.81) and tree species richness (r^2^ = 0.96) and showed negative correlation with concentration of dominance of tree (r^2^ = 0.43) (Figure 4).

Shrub density was maximum in the QL-RA community (6640 individuals ha^−1^) and minimum in the PR-QL (S) community (3786 individuals ha^−1^) (Table 1). The maximum concentration of dominance was recorded in the PR-QL (N) community (0.24), and the minimum was recorded from the QL-RA-QF community (0.11) and showed an inverse relation with Shannon diversity H′ ranged between 1.98 in PR-QL (N) community and 2.44 in the QL-RA-QF community. The maximum species diversity was recorded from PR and QL-QF-RA communities (14), and the minimum was recorded from the PR-QL community in the south (10 species). Margalef index of species richness was found to be maximum in the PR community (2.26) and minimum in the PR-QL community in the south aspect (1.09). Equitability was recorded at the maximum in the QL-RA-QF community (0.95) and minimum in the QL-RA community (0.803). Eighty-five percent of the species exhibited continuous dispersal (Table 2).

The herb density was analyzed for the three different seasons: rainy, winter and summer. During the rainy season, maximum density was recorded in the PR-QL (S) community (63.58 individuals m^−2^) and minimum in the QL-RA community (42.45 individuals m^−2^). The maximum species diversity was recorded in the PR-QL community in the north aspect (78), and the minimum was recorded in the QL-RA community (35). The QL-RA-QF community exhibited a maximum concentration of dominance (0.094), while the QL-RA community showed the lowest (0.017). Interestingly, this pattern was inversely related to Shannon diversity (H′), with the QL-RA community displaying the maximum diversity (3.99) and the QL-RA-QF community showing the lowest (2.61) (Table 3). Equitability was recorded as maximum in QL-RA-QF (0.024) and minimum in QL-RA community (0.005). Margalef species richness index was recorded as maximum in the PR community (11.58), and the minimum was recorded in the PR-QL community in the south aspect (4.90).

During the winter season, maximum density was recorded in PR-QL (S) community (9.29 individuals m^−2^) and minimum in QL-RA-QF community (3.72 individuals m^−2^). Maximum species diversity was recorded in the PR-QL community in the north aspect (16), and the minimum was recorded in the QL-RA-QF community (8). The maximum concentration of dominance was recorded in the QL-RA-QF community (0.168), and the minimum was recorded in the PR community (0.091), which showed an inverse relation with the Shannon diversity H′ (maximum was recorded in the PR community; 2.55 and minimum was recorded in QL-RA-QF community; 2.11) (Table 4). Equitability was recorded as maximum in the QL-RA-QF community (0.058) and minimum in the PR community (0.035). The maximum Margalef index of species richness was recorded in the QL-QF-RA community (2.49), and the minimum was recorded in the QL-RA-QF community (1.78).

During the summer season, maximum density was recorded in the QL-QF-RA community (40.51 individuals m^−2^) and minimum in the PR-QL (N) community (6.31 individuals m^−2^). Maximum species diversity was recorded in the QL-QF-RA community (64), and the minimum was recorded in the PR-QL community in the north aspect (11). The minimum concentration of dominance was recorded in the QL-RA community (0.046), and the maximum was recorded in the PR community (0.192). Notably, Shannon diversity (H′) followed an inverse trend, with the QL-RA community exhibiting the maximum diversity (3.27) and the PR community displaying minimum diversity (1.99) (Table 5). The maximum equitability was recorded in the PR-QL community in the north aspect (0.065) and the minimum in the QL-RA and PR-QL (S)communities (0.013). The maximum Margalef index of species richness was recorded in the QL-QF-RA community (5.47), and the minimum was recorded in the PR community (1.85).

### 2.2. Tree Biomass

Among the identified forest communities, first-year tree biomass varied from 256.8 Mg ha^−1^ to 554.2 Mg ha^−1^. Maximum biomass was recorded in the QL-RA-QF community, and minimum was recorded in PR-QL in the north aspect. The proportion of below and aboveground biomass was recorded at 22.2% and 77.8% in the PR-QL (N) community, 19.0% and 81.0% in the PR community, 22.5% and 77.5% in QL-RA community, 18.5% and 81.5% in (PR-QL) S community, 20% and 80% in QL-RA-QF community, and 21.1% and 78.9% QL-QF-RA community. The maximum biomass of the second year was recorded in the QL-RA-QF community (568.8 Mg ha^−1^), and the minimum was recorded in the PR-QL community in the north aspect (265.7 Mg ha^−1^) (Table 6). The proportion of aboveground and belowground was recorded 77.6% and 22.4% in the PR-QL (N) community, 81.1% and 18.9% inthe PR community, 77.9% and 22.1% in the QL-RA community, 81.6% and 18.4% in the (PR-QL) S community, 79.8% and 20.2% in the QL-RA-QF community, and 79.4% and 20.6% in the QL-QF-RA community. The biomass of the tree showed a strong positive correlation with the tree density (r^2^ = 0.60), total basal area of the tree (r^2^ = 0.63), tree species richness (r^2^ = 0.67) and total tree diversity (r^2^ = 0.65) (Figure 5).

### 2.3. Tree Carbon Stock and Carbon Sequestration

Among the identified forest communities, first-year tree carbon stock varied from 121.99 Mg C ha^−1^ to 263.27 Mg C ha^−1^. Maximum biomass was recorded for the QL-RA-QF community, and the minimum was recorded in PR-QL for the north aspect. In the second year, maximum biomass was recorded from the QL-RA-QF community (270.17 Mg C ha^−1^), and minimum was recorded from the PR-QL community in the north aspect (127.49 Mg C ha^−1^) (Table 7).The maximum carbon sequestered in the QL-QF-RA community (7.48 Mg C ha^−1^ yr^−1^), followed by the PR-QL (S) community (7.27 Mg C ha^−1^ yr^−1^) and QL-RA-QF (6.91 Mg C ha^−1^ yr^−1^), and the minimum was recorded for PR-QL community in south aspect (5.5 Mg C ha^−1^ yr^−1^).

### 2.4. Relationship Between Diversity and Carbon Stock

Correlation and regression analysis of carbon metrics and variables viz. basal area, density and diversity indices of trees are presented in Figure 5 and Table 8. The density of tree (r^2^ = 0.783), TBA (r^2^ = 0.790), D (r^2^ = 0.817) and SR (r^2^ = 0.824) was significantly (*p* <0.01) and positively associated with CS. The concentration of dominance of trees was observed to be significantly and negatively correlated with the C stock of trees (r^2^ = −0.756; *p* < 0.01). Tree biomass and diversity indices were also positively (*p* < 0.01) correlated.

A principal component analysis (PCA) ordination plot was constructed using eleven variables of different ecological attributes. The PCA ordination plot revealed that the first PCA axis accounted for 61.10% of the variability in species composition, while the second PCA axis explained 28.58% of this variation. According to the PCA ordination plot between PC 1 and PC 2, three forest communities (QL-RA, QL-RA-QF AND QL-QF, RA) were adjacent to each other; however, PR-QL (N), PR-QL (S) were not adjacent to each other (Figure 6). Herb density variables (e.g., RS herb density, SS herb density, WS herb density) and tree-related metrics (tree density, diversity of tree) contribute positively to both PCA 1 and PCA 2. The total basal area of the tree, the diversity of the tree, and the total carbon sequestration of the tree are clustered closely, suggesting these variables are highly correlated. PR-QL (S) is in the upper region and has a strong influence on PCA 2. QL-QF-RA and QL-RA-QF are located on the far right, indicating strong positive contributions to PCA 1.

## 3. Discussion

Bioclimatic conditions in Himalayan ecosystems fluctuate rapidly over short distances due to the highly fragmented landscape [35,36]. This variability can significantly impact vegetation types and their roles in these delicate ecosystems [8]. The current research identified 271 vascular plant species (214 herbs, 31 shrubs, and 26 trees) from 208 genera and 74 families. This finding surpasses previous studies, including those by Mandal and Joshi (2014) [37], who reported 66 species in Indian forests; Shaheen et al. (2016) [38], who documented 65 plant species in Pakistan; and Khera et al. (2001) [39], who identified 77 species in Central Himalayan forests in India. Additionally, the present study exceeds the species counts reported by Ilyas (1998) [33] from Binsar Wildlife Sanctuary (141 species), Majilla and Kala (2010) (51 species) [34], Rawat et al. (2013) [31] (147 species), Khan and Arya (2017) (23 species) [32], and Tamta (2017) (114 species) [40]. However, Wani and Pant (2023b) [41] documented a higher number of 364 species in the Gulmargh Wildlife Sanctuary. Asteraceae was the dominant family in the present study, which was similar to the findings reported by Shaheen et al. (2012) [42], Rawat et al. (2013) [31] and Singh and Pusalkar (2020) [43].

The total shrub density observed in the present study area (3786–6640 individuals ha^−1^) was higher than the total shrub density reported by Tripathi et al. (1991) [44] and Singh et al. (2014) [45] in the Kumaun Himalaya (1005–3050 individuals ha^−1^). However, it was comparable to the range of 3455–6685 individuals ha^−1^ reported by Majilla and Kala (2010) in the Binsar Wildlife Sanctuary [34]. The herb density across all three seasons, ranging from 3.72 to 63.58 individuals m^−2^, exceeded the figures reported by Tripathi et al. (1991) [44] while lower than reported by Kumar (2020) from Kalatop-Khajjiar Wildlife Sanctuary [46], Himachal Pradesh, and Wani and Pant (2023a) from Gulmargh wildlife sanctuary [25]. In the tree and shrub strata, a contagious distribution pattern was most frequently observed. The distribution pattern of woody species followed the order of contagious > random > regular, aligning with the findings of Pathak et al. (1993) for forest vegetation along an altitudinal gradient in the Kumaun Himalaya [47].

The forest communities exhibited high species diversity and low concentration of dominance for shrubs and herbs, which is characteristic of natural forests [48]. Hart and Chen (2006) emphasized that high diversity is crucial for maintaining various ecological services and regulating forest composition dynamics [49]. The concentration of dominance for shrubs in this study (0.11–0.24) was lower than that reported by Maletha et al. (2023) (0.10–0.44) in the Nanda Devi Biosphere Reserve, Western Himalaya [50], Dar and Sundarapandian (2016) (0.43–0.75) from seven temperate forest types of Western Himalaya [51] but higher (0.08–0.19) than Malik and Bhatt (2015) [52] from Kedarnath Wildlife Sanctuary. Cd of herbs of all three seasons ranged from 0.017 to 0.192, which is again lower than Dar and Sundarapandian (2016) [51] (0.08–0.35). Within the forest community, the diversity of shrubs (H′) ranged from 1.98 to 2.44, which is again lower than Dar and Sundarapandian (2016) [51] from seven temperate forest types of Western Himalaya. The herb layer, which often displays the greatest species diversity among forest strata, plays a vital role in forest biodiversity [53]. In this study, the herb layer diversity ranged from 1.99 to 3.27 for the summer season, 2.11 to 2.55 for the winter season, and 2.61 to 3.99 for the rainy season. Horvitz et al. (1998) suggested that competitive dynamics in the herbaceous layer may influence the initial success of plants in upper strata, particularly affecting the regeneration of dominant tree species [54]. In the present study, species richness for shrubs ranged from 10 to 14, which was comparable with Rana and Gairola (2009) (10–12) [55] and lower than Dar and Sundarapandian (2016) (3–9) [51]. Species richness of herbs of all three seasons ranged from 8 to 78, which was lower than Rawat et al. (2015) [56] from Nanda Devi Biosphere Reserve (18–107) and Dar and Sundarapandian (2016) [51] from temperate forests of Kashmir Himalaya (20–84). The study also revealed that the Margelf index of species richness varied across different altitudes, ranging from 1.09 to 2.61 for shrubs, 4.90–11.58 for rainy season herbs, 1.78–2.71 for winter season herbs and 1.45 to 5.47 for summer season herbs. The dominance of herbaceous plant forms observed in this research aligns with findings from various other studies conducted across different regions of the Indian Himalayan Region (IHR) [57,58,59]. This phenomenon can be attributed to the fact that herbs are typically the most common growth forms in mountainous areas, owing to their ability to adapt to diverse environmental conditions [60].

In our first-year investigations, the total biomass ranged from 256.8 Mg ha^−1^ to 554.2 Mg ha^−1^, while in the second year of investigation, the tree biomass was recorded to be in the range of 265.7 to 568.8 Mg ha^−1^. The QL-RA-QF community in the north aspect showed the maximum biomass, while PR-QL exhibited the minimum. These values were higher than those reported by Negi et al. (1983) [61]; Pandey et al. (1987) [62]; Rawat and Singh (1988) [63]; Negi et al. (1995) [64]; Gairola et al. (2010) [65]; Gossain et al. (2015) [66]; Pant and Tewari (2020) [67] and Joshi et al. (2024) [27]. However, they were lower than the figures reported by Rana et al. (1989) [68], Adhikari et al. (1995) [69], Pant and Tewari (2013) [70], Dimri et al. (2016) [71] and Karki et al. (2017) [72] from various Kumaun Himalayan forests. The possible reason for the higher biomass observed in our study, compared to other Himalayan forest studies, is the fact that the present research area is a protected wildlife sanctuary where species diversity could be on the higher side. A mature ecosystem has a significantly complex structure and trophic network, such as a natural forest or a well-established protected area. It has a substantial amount of biomass distributed among numerous taxa and individuals. Numerous factors, such as type of species, CBH, density, disturbance and climatic change, influence the contribution of different components in any given forest [73]. The proportion of below and aboveground biomass was observed to range from 18.40% to 22.25% and 77.66% to 81.02%, correspondingly, which was consistent withthe values reported for Indian forests (79% and 21%) [74]. The diversity indices and the overall tree biomass and carbon stock showed a statistically positive correlation. The positive relationship observed between overall tree biomass and carbon stock supports previous research findings [75]. Sharma et al. (2010) [76] showed a significant negative correlation between total carbon density and species diversity in their study involving twenty major Garhwal Himalayan forest types. Forests with higher tree diversity are reported to maintain better ecosystem productivity and help mitigate the effects of global warming by acting as potential carbon sinks [77]. Tree carbon stock of the present study was 127.49–269.93 Mg C/ha^−1^ and was higher than reported by Lodhiyal and Lodhiyal (2012) [78], Gossain et al. (2015) [66] and Joshi et al. (2024) [27] but lower than recorded by Jina et al. (2008) [79], Zhu et al. (2010) [80], Sharma et al. (2011) [81] and Lal and Lodhiyal (2016) [82] from natural reserved forest. The carbon storage values varied across different ranges of the study area, which could be due to variations in tree species compositions, diversity, disturbances and forest management history [83]. The carbon sequestration of the present study ranged between 5.5 and 7.48 Mg C ha^−1^ yr^−1^ and was recorded higher than reported by Jina et al. (2008) [79] and Gosain et al. (2015) [66] but lower than recorded by Rawat (1983) [84] and Rana et al. (1989) [68]. Our analysis revealed an increase in biomass and carbon density within oak-dominated communities (QL-RA-QF and QL-QF-RA) as elevation increased, indicating that high-altitude mountain ridge-top oak forests are more effective at carbon sequestration. This finding aligns with previous research supporting the notion that biomass production and carbon assimilation rise with increasing altitude [71,75,76]. At a given altitude, the combined effects of topography, aspect, slope inclination, and soil type exert significant control over plant diversity, morphometric characteristics, soil nutrient status, and community distribution [85,86]. Our research also identified key tree species that are the most common and noticeable within a specific ecosystem [87]. These prevalent species typically play a significant role in the ecosystem’s population density, stand diversity and biomass carbon storage [88,89].

## 4. Materials and Methods

### 4.1. Study Area

The Binsar Wildlife Sanctuary is a protected zone encompassing a mountain range with elevations between 1600 and 2400 m above sea level. Situated in Uttarakhand’s Kumaun region, it lies 30 km northeast of Almora district. The sanctuary spans 47.67 km^2^ and is positioned between 29°39′–29°44′ N and 79°41′–79°49′ E. (Figure 7). In 1988, this area was designated as a wildlife sanctuary to preserve and safeguard the diminishing broadleaf *Quercus* forests. Known for its exceptional natural beauty, the sanctuary boasts a unique landscape that attracts tourists. The area primarily features three major forest types: pine forests, pine–oak mixed forests, and oak forests. Located in the inner lesser Himalayan zone, the sanctuary’s geology is dominated by rocks from the Paleozoic Korlgroup, including quartzite, shale–quartzite, graphite, schist, granite and granodiorite [90]. The sanctuary is home to a diverse array of flora and fauna, including various bird species [40].

### 4.2. Phytosociological Analysis

The study was conducted from 2021 to 2023. The research area was segmented into three distinct altitudinal zones (1600–1900 m, lower altitudinal zone; 1900–2100 m, middle altitudinal zone; and 2100–2400 m, higher altitudinal zone) on both northern and southern aspects after reconnaissance survey in the region (Table 9). Within each elevation band, 100 × 100 m plots were established, with 30 randomly placed 10 × 10 m quadrats for tree sampling on both north and south-facing slopes. Tree circumference at breast height (cbh) was recorded at 1.37 m above ground level. For shrub and herb sampling, 10 quadrats of 5 × 5 m (25 m^2^) and 20 quadrats of 1 m × 1 m (1 m^2^) were nested within the same plot, adhering to standard sampling protocols [91,92]. Herbaceous species were examined across all three seasons (rainy, winter, summer). Plant specimens were identified using the relevant literature [93,94,95,96], and voucher specimens were deposited at the Department of Botany, D.S.B Campus, Kumaun University, Nainital.

In the forest zone, communities were identified following the importance value index (IVI). IVI was calculated as the sum of relative density, relative frequency and relative basal area [92]. When one species accounts for 50% or more of the total IVI, those sites are classified as pure communities of that species; when two or more species account for 50% or more of the total IVI, those sites are classified as mixed communities of those species. The total number of species in a given community was used to calculate species richness using the Margalef index [97]. Simpson’s Index [98] was used to calculate the concentration of dominance, and Shannon–Wiener’s information statistic (H′) [99] was used to determine species diversity and to determine species evenness; the method given by Magurran (1988) was used [100].

### 4.3. Biomass Estimation

To evaluate annual diameter growth, all measured trees were marked with yellow paint at breast height (1.37 m) in September 2021 (B_1_). The equation y = a + blnx was employed, where y represents the component’s dry weight (kg), x is the CBH (cm), a denotes the intercept, b signifies the regression coefficient, and ln stands for natural logarithm. Component-specific biomass (above ground—bole, bark, branch, twigs, foliage; below ground—stump root, lateral root and fine root) was determined using allometric equations of Chaturvedi and Singh (1983) [101], Adhikari et al. (1995) [69], Adhikari et al. (1998) [102] and Rawat and Singh (1988) [63]. Following CBH re-measurement in September 2022 (B_2_), the two-year net biomass change was calculated. Annual biomass accumulation was defined as the change in biomass (ΔB = B_2_ − B_1_), with B_2_ and B_1_ representing second and first-year biomass, respectively. The sum of ΔB values across components yielded the trees’ net biomass accretion.

### 4.4. Carbon Stock and Carbon Sequestration

Carbon stock was calculated by multiplying species biomass by 0.475, as per Magnuseen and Reed (2004) [103]. Initial carbon stock (C_1_) was determined for each forest in 2021. After one year, the increased carbon stock (C_2_) was estimated using the enhanced biomass value for each stand/site. The net change in carbon stock (ΔC = C_2_ − C_1_) provided the annual carbon accumulation.

### 4.5. Statistical Analysis of the Data

Regression analysis (linear) was performed in Microsoft Excel Windows11 to ascertain the association between the density of trees and other ecological attributes. At the 95% significance level, the *p* values were tested at less than 0.05 (significant) and 0.01 (highly significant). Using Past software (version 3), the Pearson correlation analysis was per formed to determine the linear correlation (significant, *p* < 0.05) between the various ecological attributes. Principal component analysis (PCA) was performed to summarize the compositional differences between various sites and was analyzed using PAST software [104].

## 5. Conclusions

The current study’s results highlighted the significance of wildlife sanctuaries, which not only aid in wildlife preservation but also play a substantial role in carbon sequestration and support a wider range of biodiversity. Temperate forests account for greater biomass and C stock than subtropical forests. Research findings indicated that forests dominated by *Quercus* species contained the highest levels of biomass and carbon storage within protected areas, which is particularly relevant given current climate change concerns. We found that structural attributes governed the biomass carbon stock in selected forest communities, mainly tree density, total tree basal area, tree species richness and overall tree diversity. Elevation showed significant positive correlations as, with increasing altitude, the biomass and carbon stock increased, indicating that the entire elevational range has a huge carbon stock. Furthermore, few dominant species with large diameters account for the majority of the C stock of these forests. Hence, the cutting and felling of these species must be regulated for long-term ecosystem sustainability. Given the ecological importance of oak species, it is advised to utilize smaller trees or those outside forested areas for fuel and timber purposes. To ensure the future stability of the carbon supply, these species must be used responsibly and allowed to regenerate naturally. Protecting oak species is crucial for effective land management, as they constitute the majority of the area’s carbon stock and sequestration potential. To reduce the pressure on natural forests from human activities, fast-growing, high-density tree species or fuel wood plantations could be established along village corridors. This kind of dynamic study of carbon storage will help develop effective environmental management strategies to mitigate the effects of climate change, improve conservation efforts and support the sustainable utilization of forest resources. Afforestation programs using the Assisted Natural Regeneration (ANR) technique should be initiated in the forests. These afforestation programs could also contribute to the Paris Climate Agreement, the Trillion Trees initiative, and the Bonn Challenge, which aims to restore 350 million ha of degraded and deforested areas by 2030.

### Implications for Climate Change Mitigation

Forest management sustainability on a global scale was evaluated using two key indicators: species diversity and carbon. However, forest managers may face challenges in simultaneously preserving diversity and enhancing carbon storage. These factors have a significant impact that goes beyond their physical presence, affecting the overall ecological balance of forest environments. The present study emphasizes the need to implement multispecies and multi-layered eco-restoration strategies, prioritizing the inclusion of shrubs and herbs alongside tree species to maintain overall biodiversity and ensure the continued delivery of ecosystem services and goods. Although tree biomass and carbon stock contributed 554.2 Mg ha^−1^ and 270.17 Mg C ha^−1^, respectively, they are crucial for maintaining the ecological stability and functionality of forest ecosystems. Consequently, biomass must be recognized as a critical component of forest ecosystems and preserved accordingly. In the BWLS landscapes, the *Quercus*-dominated forest type should be prioritized for protection and restoration due to its higher carbon content compared to other forest types. Given the increasing focus on global ecological restoration, forest vegetation should be considered a fundamental element in forest conservation and restoration efforts. The planting of broadleaved trees should be promoted in the area, as they enhance forest cover, mitigate soil erosion, sustain soil fertility, improve water infiltration and aeration, and enrich soil nutrients.

## Figures and Tables

**Figure 1 plants-14-00291-f001:**
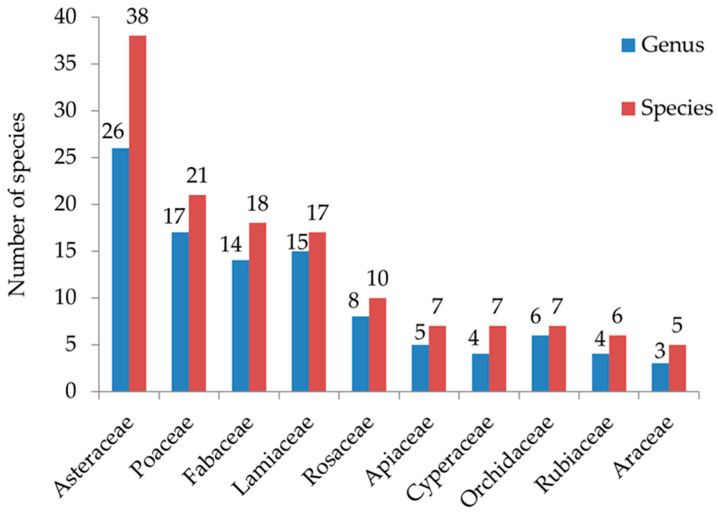
Dominant families with number of taxa in Binsar Wildlife Sanctuary.

**Figure 2 plants-14-00291-f002:**
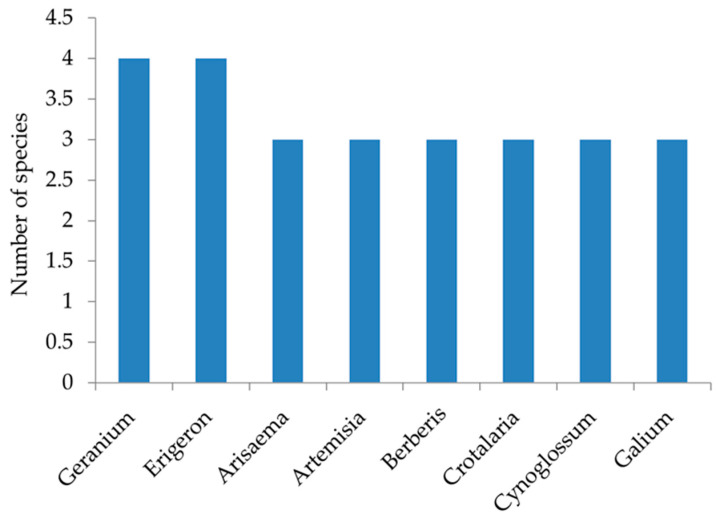
Dominant genera in Binsar Wildlife Sanctuary.

**Figure 3 plants-14-00291-f003:**
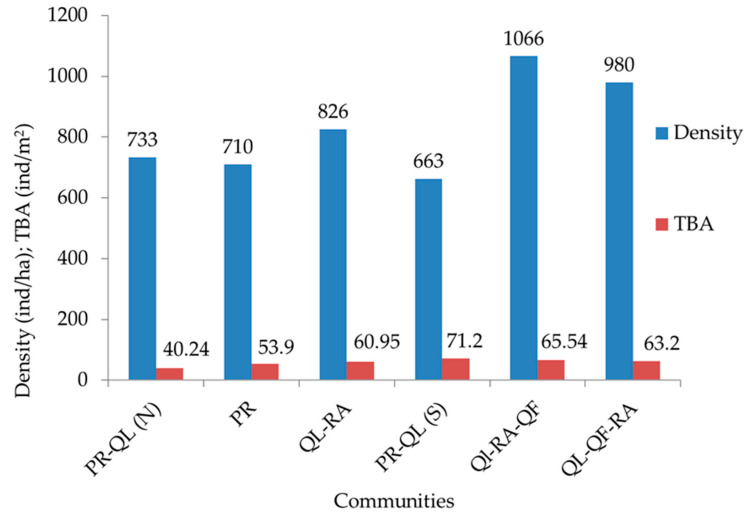
Density (individuals ha^−1^) and total basal area (m^2^ h^−2^) of trees in BWLS.

**Figure 4 plants-14-00291-f004:**
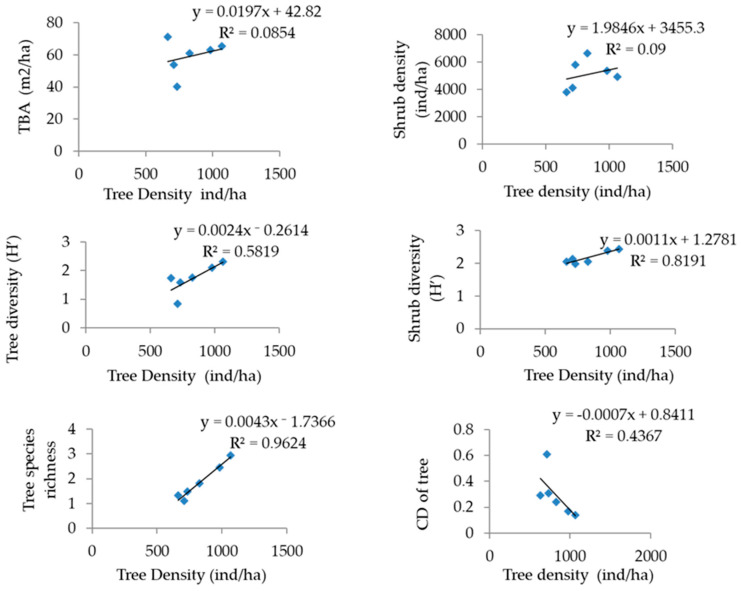
Correlation of tree density with total basal area of tree; tree diversity; shrub density; shrub diversity; tree species richness and concentration of dominance of trees.

**Figure 5 plants-14-00291-f005:**
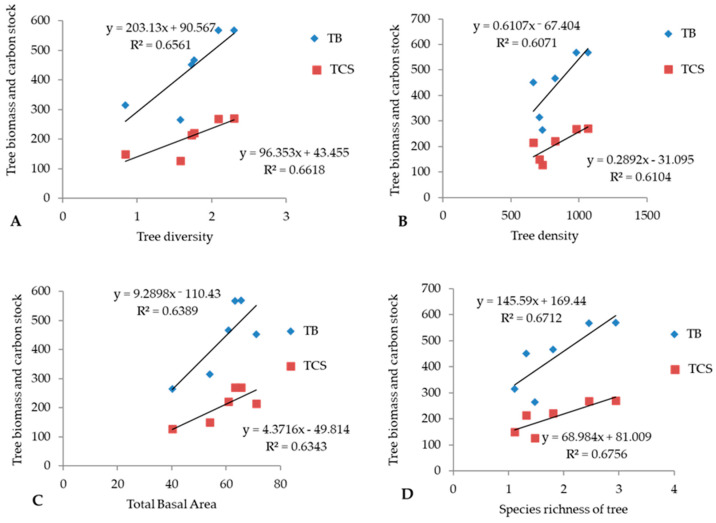
Correlation of biomass and carbon stock: (**A**) tree diversity; (**B**) tree density; (**C**) total basal area; (**D**) species richness in the BWLS.

**Figure 6 plants-14-00291-f006:**
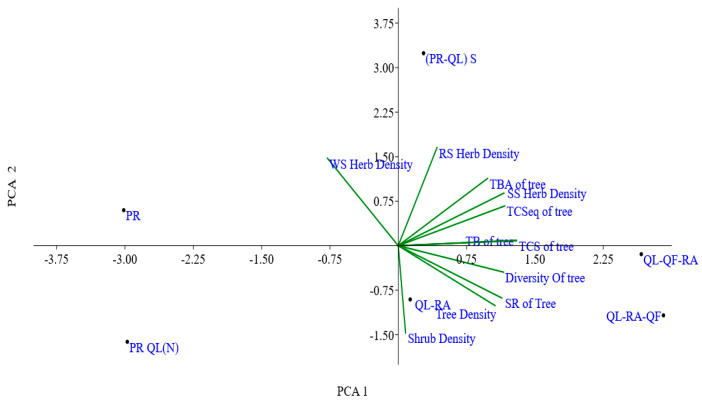
Principal component analysis (PCA) based on different ecological attributes of six forest communities. Abbreviations used: (RS: rainy season; WS: winter season; SS: summer season; TCSeq: total carbon sequestration; SR: species richness; TCS: total carbon stock).

**Figure 7 plants-14-00291-f007:**
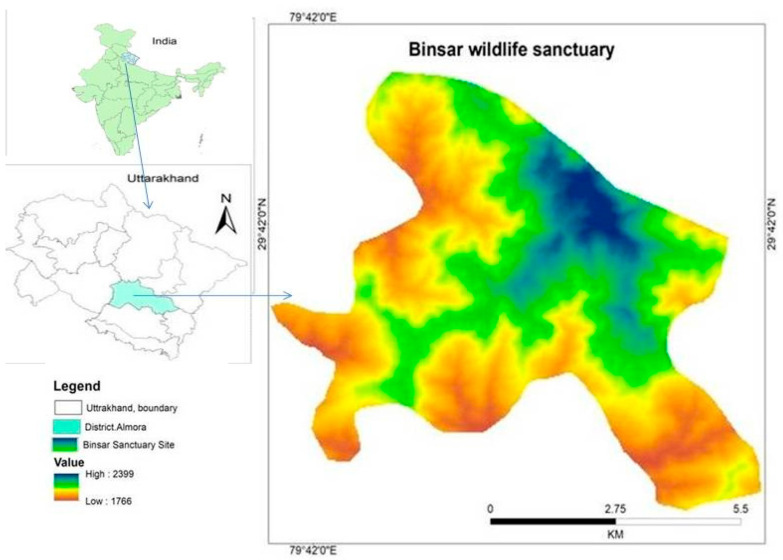
Map of the study area.

**Table 1 plants-14-00291-t001:** Community-wise density of shrubs and herbs in BWLS.

	PR-L (N)	PR	QL-RA	PR-QL (S)	QL-RA-QF	QL-QF-RA
**Density (individuals ha^−1^)**						
Shrub	5813	4106	6640	3786	4920	5346
**Density (individuals m^−2^)**						
Rainy season herb	42.99	44.16	42.45	63.58	44.45	53.94
Winter season herb	6.01	8.2	6.47	9.29	3.72	4.52
Summer season herb	6.31	15.81	22.33	40.30	37.95	40.51

**Table 2 plants-14-00291-t002:** Community-wise diversity–dominance indexes of shrub in the BWLS m^2^.

Communities	Taxa	Dominance	Diversity	Evenness	Species Richness
PR-QL (N)	11	0.240	1.98	0.82	1.15
PR	14	0.156	2.13	0.807	2.26
QL-RA	13	0.173	2.06	0.803	1.36
(PR-QL) S	10	0.220	2.05	0.89	1.09
QL-RA-QF	13	0.110	2.44	0.95	1.41
QL-QF-RA	14	0.120	2.39	0.90	1.51

**Table 3 plants-14-00291-t003:** Community-wise diversity–dominance indexes of rainy season herb in BWLS.

Communities	Taxa	Diversity	Dominance	Evenness	Species Richness
PR-QL (N)	78	3.38	0.034	0.008	6.45
PR	50	3.65	0.023	0.006	11.58
QL-RA	35	3.99	0.017	0.005	9.15
(PR-QL) S	70	3.34	0.047	0.011	4.90
QL-RA-QF	50	2.61	0.094	0.024	6.75
QL-QF-RA	51	3.28	0.049	0.012	6.71

**Table 4 plants-14-00291-t004:** Community-wise diversity–dominance indexes of winter season herb in BWLS.

Communities	Taxa	Diversity	Dominance	Evenness	Species Richness
PR-QL (N)	16	2.21	0.160	0.058	2.09
PR	13	2.55	0.091	0.035	2.71
QL-RA	15	2.25	0.147	0.054	2.05
(PR-QL) S	13	2.49	0.096	0.037	2.51
QL-RA-QF	8	2.11	0.168	0.081	1.78
QL-QF-RA	13	2.46	0.098	0.038	2.49

**Table 5 plants-14-00291-t005:** Community-wise diversity–dominance indexes of summer season herb in BWLS.

Communities	Taxa	Diversity	Dominance	Evenness	Species Richness
PR-QL (N)	11	2.81	0.156	0.065	3.38
PR	23	1.99	0.192	0.061	1.85
QL-RA	34	3.27	0.046	0.013	4.59
(PR-QL) S	38	3.25	0.049	0.013	4.41
QL-RA-QF	57	3.07	0.087	0.022	5.31
QL-QF-RA	64	2.84	0.095	0.023	5.47

**Table 6 plants-14-00291-t006:** Community-wise first-year and second-year biomass (Mg ha^−1^) of trees BWLS.

	First-Year Biomass			Second-Year Biomass		
Communities	AGB	BGB	TB	AGB	BGB	TB
PR-QL (N)	199.7	57.1	256.8	205.7	60.0	265.7
PR	245.2	57.7	302.9	255.2	59.5	314.7
QL-RA	349.7	101.6	451.3	363.4	103.2	466.6
(PR-QL) S	356.0	81.0	4370	369.0	83.2	452.2
QL-RA-QF	443.1	111.1	554.2	454.1	114.7	568.8
QL-QF-RA	435.6	116.5	552.1	450.5	117.1	567.6

Abbreviations used: AGB: aboveground biomass; BGB: belowground biomass; TB: total biomass.

**Table 7 plants-14-00291-t007:** Community-wise carbon stock (Mg C ha^−1^) and carbon sequestration (Mg ha^−1^ yr^−1^) of trees in BWLS.

	First-Year Carbon Stock			Second-Year Carbon Stock			
Communities	AGCS	BGCS	TCS	AGCS	BGCS	TCS	TCSeq
PR-QL (N)	94.88	27.12	121.99	99.51	27.98	127.49	5.50
PR	116.45	27.41	143.86	121.23	28.28	149.51	5.65
QL-RA	166.37	48.24	214.61	172.46	49.00	221.46	6.85
(PR-QL) S	169.10	38.46	207.56	175.27	39.53	214.80	7.24
QL-RA-QF	210.49	52.78	263.27	215.70	54.47	270.17	6.91
QL-QF-RA	206.90	55.35	262.25	214.00	55.73	269.73	7.48

Abbreviations used: AGB: aboveground carbon stock; BGB: belowground carbon stock; TB: total carbon stock; TCSeq: total carbon sequestration.

**Table 8 plants-14-00291-t008:** Pearson’s correlation analysis among different variables.

	Density	TBA	D	CD	SR	B	CS
Density	1						
TBA	0.292	1					
D	0.762 **	0.462	1				
CD	−0.688 *	−0.392	−0.980 **	1			
SR	0.980 **	0.358	0.867 **	−0.799 **	1		
B	0.781 **	0.794 **	0.813 **	−0.751 **	0.821 **	1	
CS	0.783 **	0.790 **	0.817 **	−0.756 **	0.824 **	0.999 *	1

**: Significant at <1% level of significance; *: significant at <5% level of significance; TBA: total basal area of tree, D: diversity of tree, CD: concentration of dominance of tree, SR: species richness of trees; B: biomass of trees, CS: carbon stock of trees.

**Table 9 plants-14-00291-t009:** Location and characteristics of studied sites.

Forest Zone	Altitude (m a.s.l)	Aspect	Dominant Tree Species	Associated Herb and Shrub
PR-QL (N)	1600–1900 (Lower altitudinal zone)	North	*Pinus roxburghii*, *Quercus leucotrichophora*	*Berberis asiatica*, *Myrsine africana*, *Vincetoxicum glaucum*, *Achyranthes bidentata*
PR	1600–1900 (Lower altitudinal zone)	South	*Pinus roxburghii*	*Rubus ellipticus*, *Glochidion heyneanum*, *Ageratina adenophora*, *Theropogon pallidus*
QL-RA	1900–2100 (Middle altitudinal zone)	North	*Quercus leucotrichophora*, *Rhododendron arboreum*	*Daphne papyracea*, *Coriaria napalensis*, *Dicliptera bupleuroides*, *Strobilanthes atropurpurea*
PR-QL (S)	1900–2100 (Middle altitudinal zone)	South	*Pinus roxburghii*, *Quercus leucotrichophora*	*Viburnum cotinifolium*, *Ototropis multiflora*, *Rostellularia diffusa, Pimpinella diversifolia*
QL-RA-QF	2100–2400 (Higher altitudinal zone)	North	*Quercus leucotrichophora*, *Rhododendron arboreum*, *Quercus floribunda*	*Berberis aristata*, *Drepanostachyum falcatum*, *Strobilanthes atropurpurea*, *Valeriana hardwickei*
QL-QF-RA	2100–2400 (Higher altitudinal zone)	South	*Quercus leucotrichophora*, *Quecus floribunda*, *Rhododendron arboreum*	*Myrsine africana*, *Berberis napaulensis*, *Stellaria patens*, *Origanum vulgare*

Abbreviation used: PR: *Pinus roxburghii*; QL: *Quercus leucotrichophora*; RA: *Rhododendron arboreum*; QF: *Quercus floribunda*; N: north; S: south.

## Data Availability

The raw data supporting the conclusions of this article will be made available by the authors on request.

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
