# Peer review of "Species Diversity, Biomass Production and Carbon Sequestration Potential in the Protected Area of Uttarakhand, India"

_plants, 2025, doi:10.3390/plants14020291_

Round 1
Reviewer 1 Report
Comments and Suggestions for Authors
Overall Assessment
This study investigates the diversity of forest understory (shrubs and herbs), as well as tree biomass and carbon sequestration, in the Binsar Wildlife Reserve, India. The findings highlight that forests dominated by Quercus species possess the highest levels of biomass and carbon storage within the study area. The study provides strong evidence in favour of forest management strategies that prioritize the integration of multiple vegetation layers in landscape restoration, enhancing biodiversity and improving resilience to climate change.
However, the study has several shortcomings in its description and logic, and the English language needs refinement. The study is recommended for publication, but only after major revisions.
Abstract
The abstract is overly detailed regarding the results, while the overall context is limited to just one brief sentence. Consider balancing these aspects by providing a more comprehensive context and summarizing the findings succinctly.
Introduction
The literature review is too brief and lacks specificity. The section on the study area should be moved to the Materials and Methods section for better organization.
For Figure 1, adding a map of India would help readers locate the study region more easily.
Materials and Methods
• Sampling and biomass estimation are clearly described.
• The method for estimating below-ground biomass is unclear—was it not included? Clarify this aspect of carbon stock estimation.
• Statistical analysis is entirely missing from this section, despite the results and discussion referencing several correlation analyses. A detailed explanation of the statistical methods used should be included here.
Results
• Results are described in detail.
• Figure 3 uses a different font compared to other figures. Ensure font consistency across all figures.
• The term "Community-wise" should be correctly spelled in all figure captions and throughout the text.
• Numerous statistical results are presented; ensure that the corresponding methods are thoroughly detailed in the Materials and Methods section.
Discussion
• Figure 7 should be moved to the Results section as it presents results.
• The discussion is thorough and detailed.
Conclusions
The conclusions are concise, but a discussion on future directions or a "way forward" should be included to enhance the impact of this section.
The English language needs refinement.
Author Response
Clarifications to the Comments and Suggestions
Manuscript id: plants-3396608
Manuscript title: Species Diversity, Biomass Production and Carbon Sequestration Potential in the Protected Area of Uttarakhand, India
Reviewer 1
Comment: The abstract is overly detailed regarding the results, while the overall context is limited to just one brief sentence. Consider balancing these aspects by providing a more comprehensive context and summarizing the findings succinctly.
Explanation: As per the suggestion by the reviewer, the abstract has been modified in the revised manuscript.
Comment: The literature review is too brief and lacks specificity.
Explanation: As per the suggestion by the reviewer, the literature review has been elaborated in the revised manuscript.
Comment: The section on the study area should be moved to the Materials and Methods section for better organization.
Explanation: As per the suggestion by the reviewer, the study area moved to the Materials Methods section of the revised manuscript.
Comment: For Figure 1, adding a map of India would help readers locate the study region more easily.
Explanation: As per the suggestion by reviewer, map of India has been included in the Figure 1 of the revised manuscript.
Comment: The method for estimating below-ground biomass is unclear-was it not included? Clarify this aspect of carbon stock estimation.
Explanation: For calculating Below ground biomass, stump root, lateral root and fine root were taken into account. This has been clearly mentioned in the Materials and Methods section of the revised manuscript. Below-ground biomass was included in the manuscript. After calculating the carbon stock of the two consecutive years (C1= Carbon stock Ist year; C2= Carbon stock IInd year), the change in the carbon stock C2-C1 yields the carbon sequestration.
Comment: Statistical analysis is entirely missing from this section, despite the results and discussion referencing several correlation analyses. A detailed explanation of the statistical methods used should be included here.
Explanation: As per the suggestion, statistical analysis was included at the appropriate place in the Materials and Methods section of the revised manuscript.
Comment: Figure 3 uses a different font compared to other figures. Ensure font consistency across all figures.
Explanation: As per the suggestion, the font consistency was corrected in the revised manuscript.
Comment: The term "Community-wise" should be correctly spelled in all figure captions and throughout the text.
Explanation: As per the suggestion by reviewer, community-wise correctly spelled in the Figures of the revised manuscript.
Comment: Numerous statistical results are presented; ensure that the corresponding methods are thoroughly detailed in the Materials and Methods section.
Explanation: The needful has been done in the revised manuscript.
Comment: The conclusions are concise, but a discussion on future directions or a "way forward" should be included to enhance the impact of this section.
Explanation: As the suggestion by the reviewer, conclusions section has been modified in the revised manuscript.
Comment: Figure 7 should be moved to the Results section as it presents results
Explanation: As per the suggestion by the reviewer, Figure 7 has been included in the Results section of the revised manuscript.
Reviewer 2
Comment: My first major concern is that the authors do not clearly state what is the hypothesis or mechanism being tested or addressed in their study.
Explanation: As per the suggestion given by the reviewer, the hypothesis, or mechanism being tested was addressed in the study in the revised manuscript.
Comment: While the information collected is certainly very interesting, it is not clear what is the underlying rationale for the study's sampling design. I found it very startling that the authors focus on the phytosociological classes identified when comparing the various response variables but made no mention of the potential causes such as aspect or elevation.
Explanation: The potential causes such as aspect or elevation on the various response variables have been added in the Discussion part of the revised manuscript (line 392-396) along with the references.
Comment: Indeed, the closing paragraph of the introduction makes mention of the earlier studies of Joshi et al. (2024) but fails to mention other references mentioned later by the authors in the discussion (e.g. lines 309 onwards).
Explanation: The study of Joshi et al. (2024) is from wildlife centaury while the references mentioned in the discussion part (e.g. lines 309 onwards), have been done on the Himalayan forest other than wild life Sanctuary. The references mentioned in the para “Previous studies in the Binsar Wildlife Sanctuary primarily focused on the phytosociological characteristics of tree species [27, 31, 32]. Limited research has been conducted on other life forms such as shrubs and herbs [33, 34]. Furthermore, Joshi et al. (2024) have examined carbon stock of the trees in this study site [27].” Investigated different aspects in the Binsar Wildlife sanctuary. Therefore, the references mentioned in the discussion part, have not included in the closing paragraph of introduction.
Comment: This lack of underlying rationale or hypothetical overarching mechanism carries over to the result section, which focuses on listing the range of variability observed in the various metrics studied, which yields no insight, as no systematic comparison or pattern emerges. As a result, most of the results are very confusing, and very difficult to follow. For instance, from lines 161 to 211 the authors list the maxima and minima observed for all the response variables. However, this information, which is already contained in Tables 2 to 6, is not organized regarding some reference causal variable. Given that the two first candidate variables (elevation and aspect) have been obscured by associating them with a phytosociological grouping, it is difficult to discern any underlying pattern.
Explanation: The results section has been modified as per the suggestions of the reviewer in the manuscript.
Comment: In addition, no formal statistical comparison is made across the different phytosociological groups, and none of the reported values show a measure of variability (standard error or standard deviation). This makes it very difficult to interpret the large amount of valuable information collected by the authors.
Explanation: Formal statistical comparison was not made across different phytosociological groups. Within each elevation band, 100×100 m plots were established, with 30 randomly placed 10×10 m quadrats for tree sampling on both north and south-facing slopes. In our study, regression analysis (linear) was performed to ascertain the association between the density of trees and other ecological attributes. Pearson correlation analysis was done to determine the liner correlation between the various ecological attributes. Principal Component Analysis (PCA) was performed to summarize the compositional differences between various sites.
Most of such kind of study did not include statistical comparison in the different phytosociological groups. Statistical comparison was included in the soil profile data in those studies. Some of the references have been given for support.
- Thapliyal et al., 2024 https://doi.org/10.1016/j.tfp.2024.100690
- Bisht, S.; Bargali, S.S.; Bargali, K.; Rawat, G.S.; Rawat, Y.S.; Fartyal, A. Influence of Anthropogenic Activities on Forest Carbon Stocks-A Case Study from Gori Valley, Western Himalaya. Sustainability 2022, 14, 16918. https://doi.org/10.3390/su142416918
Comment: A second concern is the structure and format of graphical and tabular information. As mentioned above, none of the tables include measures of variability. The figures differ in style and format: (i) The font used in Figure 4 differs from the other Figures. (ii) Figures 2 to 6 use bounding frames or boxes, while Figure 7 does not. (iii) Figure 6 is the only one to label subplot, using an awkward format (lower left-hand corner), (iv) Several of the axes do not show any axis titles (e.g. Figs 1, 2, 3, 4), (v) The axis labels in Figure are not uniform in style or unit labelling.
Explanation: As per the suggestion by reviewer, the figures style and format have been corrected in the revised manuscript.
Comment: I suggest the authors frame their results in the corresponding theoretical framework, namely, how their results provide insights on the role or interplay of functional diversity and carbon stock or storage. I believe the available information may be presented in a very informative manner if the authors explicitly compare or relate the carbon storage ability of various forest types with the functional diversity of taxa present. A revised version of the analysis presented in Figures 5 and 6 may allow the authors to draw very interesting conclusions regarding the relationship between facets of diversity and carbon stock. Focusing on a clear rationale for the study will allow the authors to structure and present their findings more effectively.
Explanation: The role of functional diversity on carbons stock has been included in the revised manuscript as per the reviewer’s suggestions in the abstract (line no. 30-33), results section (line no. 282-287), discussion section (Line no. 389-394) and conclusion part (line no. 445-447)
All the changes have been incorporated in the revised manuscript and the corrections are made by yellow highlights. We express our heartful gratitude to the reviewers for their valuable suggestions and corrections, which have helped us in improving the quality of this manuscript.
Yours Sincerely,
R.K. Chaturvedi
Reviewer 2 Report
Comments and Suggestions for Authors
The study by Geetanjali Upadhyay and colleagues aims to quantify forest understory (shrub and herb) diversity, tree biomass, and carbon sequestration potential of the forests in the Binsar Wildlife sanctuary (BWLS), Uttarakhand, India. The study builds on a phytosociological study conducted between 2021 and 2023, within the BWLS. The study addresses a set of relevant research questions. However, I have some major comments:
My first major concern is that the authors do not clearly state what is the hypothesis or mechanism being tested or addressed in their study. While the information collected is certainly very interesting, it is not clear what is the underlying rationale for the study's sampling design. I found it very startling that the authors focus on the phytosociological classes identified when comparing the various response variables but made no mention of the potential causes such as aspect or elevation. Indeed, the closing paragraph of the introduction makes mention of the earlier studies of Joshi et al. (2024) but fails to mention other references mentioned later by the authors in the discussion (e.g. lines 309 onwards). This lack of underlying rationale or hypothetical overarching mechanism carries over to the result section, which focuses on listing the range of variability observed in the various metrics studied, which yields no insight, as no systematic comparison or pattern emerges. As a result, most of the results are very confusing, and very difficult to follow. For instance, from lines 161 to 211 the authors list the maxima and minima observed for all the response variables. However, this information, which is already contained in Tables 2 to 6, is not organized regarding some reference causal variable. Given that the two first candidate variables (elevation and aspect) have been obscured by associating them with a phytosociological grouping, it is difficult to discern any underlying pattern. In addition, no formal statistical comparison is made across the different phytosociological groups, and none of the reported values show a measure of variability (standard error or standard deviation). This makes it very difficult to interpret the large amount of valuable information collected by the authors.
A second concern is the structure and format of graphical and tabular information. As mentioned above, none of the tables include measures of variability. The figures differ in style and format: (i) The font used in Figure 4 differs from the other Figures. (ii) Figures 2 to 6 use bounding frames or boxes, while Figure 7 does not. (iii) Figure 6 is the only one to label subplot, using an awkward format (lower left-hand corner), (iv) Several of the axes do not show any axis titles (e.g. Figs 1, 2, 3, 4), (v) The axis labels in Figure are not uniform in style or unit labelling.
Finally, given that the study has been submitted to the Special Issue "Plant Functional Diversity and Nutrient Cycling in Forest Ecosystems", I suggest the authors frame their results in the corresponding theoretical framework, namely, how their results provide insights on the role or interplay of functional diversity and carbon stock or storage. I believe the available information may be presented in a very informative manner if the authors explicitly compare or relate the carbon storage ability of various forest types with the functional diversity of taxa present. A revised version of the analysis presented in Figures 5 and 6 may allow the authors to draw very interesting conclusions regarding the relationship between facets of diversity and carbon stock. Focusing on a clear rationale for the study will allow the authors to structure and present their findings more effectively.
Author Response
Reviewer 2
Comment: My first major concern is that the authors do not clearly state what is the hypothesis or mechanism being tested or addressed in their study.
Explanation: As per the suggestion given by the reviewer, the hypothesis, or mechanism being tested was addressed in the study in the revised manuscript.
Comment: While the information collected is certainly very interesting, it is not clear what is the underlying rationale for the study's sampling design. I found it very startling that the authors focus on the phytosociological classes identified when comparing the various response variables but made no mention of the potential causes such as aspect or elevation.
Explanation: The potential causes such as aspect or elevation on the various response variables have been added in the Discussion part of the revised manuscript (line 392-396) along with the references.
Comment: Indeed, the closing paragraph of the introduction makes mention of the earlier studies of Joshi et al. (2024) but fails to mention other references mentioned later by the authors in the discussion (e.g. lines 309 onwards).
Explanation: The study of Joshi et al. (2024) is from wildlife centaury while the references mentioned in the discussion part (e.g. lines 309 onwards), have been done on the Himalayan forest other than wild life Sanctuary. The references mentioned in the para “Previous studies in the Binsar Wildlife Sanctuary primarily focused on the phytosociological characteristics of tree species [27, 31, 32]. Limited research has been conducted on other life forms such as shrubs and herbs [33, 34]. Furthermore, Joshi et al. (2024) have examined carbon stock of the trees in this study site [27].” Investigated different aspects in the Binsar Wildlife sanctuary. Therefore, the references mentioned in the discussion part, have not included in the closing paragraph of introduction.
Comment: This lack of underlying rationale or hypothetical overarching mechanism carries over to the result section, which focuses on listing the range of variability observed in the various metrics studied, which yields no insight, as no systematic comparison or pattern emerges. As a result, most of the results are very confusing, and very difficult to follow. For instance, from lines 161 to 211 the authors list the maxima and minima observed for all the response variables. However, this information, which is already contained in Tables 2 to 6, is not organized regarding some reference causal variable. Given that the two first candidate variables (elevation and aspect) have been obscured by associating them with a phytosociological grouping, it is difficult to discern any underlying pattern.
Explanation: The results section has been modified as per the suggestions of the reviewer in the manuscript.
Comment: In addition, no formal statistical comparison is made across the different phytosociological groups, and none of the reported values show a measure of variability (standard error or standard deviation). This makes it very difficult to interpret the large amount of valuable information collected by the authors.
Explanation: Formal statistical comparison was not made across different phytosociological groups. Within each elevation band, 100×100 m plots were established, with 30 randomly placed 10×10 m quadrats for tree sampling on both north and south-facing slopes. In our study, regression analysis (linear) was performed to ascertain the association between the density of trees and other ecological attributes. Pearson correlation analysis was done to determine the liner correlation between the various ecological attributes. Principal Component Analysis (PCA) was performed to summarize the compositional differences between various sites.
Most of such kind of study did not include statistical comparison in the different phytosociological groups. Statistical comparison was included in the soil profile data in those studies. Some of the references have been given for support.
- Thapliyal et al., 2024 https://doi.org/10.1016/j.tfp.2024.100690
- Bisht, S.; Bargali, S.S.; Bargali, K.; Rawat, G.S.; Rawat, Y.S.; Fartyal, A. Influence of Anthropogenic Activities on Forest Carbon Stocks-A Case Study from Gori Valley, Western Himalaya. Sustainability 2022, 14, 16918. https://doi.org/10.3390/su142416918
Comment: A second concern is the structure and format of graphical and tabular information. As mentioned above, none of the tables include measures of variability. The figures differ in style and format: (i) The font used in Figure 4 differs from the other Figures. (ii) Figures 2 to 6 use bounding frames or boxes, while Figure 7 does not. (iii) Figure 6 is the only one to label subplot, using an awkward format (lower left-hand corner), (iv) Several of the axes do not show any axis titles (e.g. Figs 1, 2, 3, 4), (v) The axis labels in Figure are not uniform in style or unit labelling.
Explanation: As per the suggestion by reviewer, the figures style and format have been corrected in the revised manuscript.
Comment: I suggest the authors frame their results in the corresponding theoretical framework, namely, how their results provide insights on the role or interplay of functional diversity and carbon stock or storage. I believe the available information may be presented in a very informative manner if the authors explicitly compare or relate the carbon storage ability of various forest types with the functional diversity of taxa present. A revised version of the analysis presented in Figures 5 and 6 may allow the authors to draw very interesting conclusions regarding the relationship between facets of diversity and carbon stock. Focusing on a clear rationale for the study will allow the authors to structure and present their findings more effectively.
Explanation: The role of functional diversity on carbons stock has been included in the revised manuscript as per the reviewer’s suggestions in the abstract (line no. 30-33), results section (line no. 282-287), discussion section (Line no. 389-394) and conclusion part (line no. 445-447)
All the changes have been incorporated in the revised manuscript and the corrections are made by yellow highlights. We express our heartful gratitude to the reviewers for their valuable suggestions and corrections, which have helped us in improving the quality of this manuscript.
Yours Sincerely,
R.K. Chaturvedi
Round 2
Reviewer 1 Report
Comments and Suggestions for Authors
The authors have made substantial improvements to the manuscript, making it suitable for publication. However, further refinement of the English language is necessary prior to publication.
Comments on the Quality of English LanguageFurther refinement of the English language is necessary prior to publication.
Reviewer 2 Report
Comments and Suggestions for Authors
The authors have addressed the majority of the observations I indicated in my previous review report. I commend the authors for their efforts and appreciate their detailed responses.